# Production of a Series of Long-Chain Isomaltooligosaccharides from Maltose by *Bacillus subtilis* AP-1 and Associated Prebiotic Properties

**DOI:** 10.3390/foods12071499

**Published:** 2023-04-03

**Authors:** Suratsawadee Tiangpook, Sreyneang Nhim, Pattaneeya Prangthip, Patthra Pason, Chakrit Tachaapaikoon, Khanok Ratanakhanokchai, Rattiya Waeonukul

**Affiliations:** 1Division of Biochemical Technology, School of Bioresources and Technology, King Mongkut’s University of Technology Thonburi (KMUTT), Bangkok 10150, Thailand; 2Department of Tropical Nutrition & Food Science, Faculty of Tropical Medicine, Mahidol University, Bangkok 10400, Thailand; 3Excellent Center of Enzyme Technology and Microbial Utilization, Pilot Plant Development and Training Institute (PDTI), King Mongkut’s University of Technology Thonburi (KMUTT), Bangkok 10150, Thailand

**Keywords:** *Bacillus subtilis*, α-glucosidase, isomaltooligosaccharide, prebiotic, transglucosidase

## Abstract

*Bacillus subtilis* strain AP-1, which produces α-glucosidase with transglucosidase activity, was used to produce a series of long-chain isomaltooligosaccharides (IMOs) with degree of polymerization (DP) ranging from 2 to 14 by direct fermentation of maltose. A total IMOs yield of 36.33 g/L without contabacillusmination from glucose and maltose was achieved at 36 h of cultivation using 50 g/L of maltose, with a yield of 72.7%. IMOs were purified by size exclusion chromatography with a Superdex 30 Increase column. The molecular mass and DP of IMOs were analyzed by matrix-assisted laser desorption/ionization time-of-flight mass spectrometry (MALDI-TOF/MS). Subsequently, linkages in produced oligosaccharides were verified by enzymatic hydrolysis with α-amylase and oligo-α-1,6-glucosidase. These IMOs showed prebiotic properties, namely tolerance to acidic conditions and digestive enzymes of the gastrointestinal tract, stimulation of probiotic bacteria growth to produce short-chain fatty acids and no stimulating effect on pathogenic bacteria growth. Moreover, these IMOs were not toxic to mammalian cells at up to 5 mg/mL, indicating their biocompatibility. Therefore, this research demonstrated a simple and economical method for producing IMOs with DP2–14 without additional operations; moreover, the excellent prebiotic properties of the IMOs offer great prospects for their application in functional foods.

## 1. Introduction

Prebiotics are non-digestible food ingredients that provide beneficial effects on the host’s health by promoting a good balance of intestinal microflora in the gut and decreasing gastrointestinal infections [1,2]. Isomaltooligosaccharides (IMOs), one of the most promising prebiotics among all the oligosaccharides, are glucosyl saccharides containing α-D-(1,6) linkages, with or without α-D-(1,4), (1,3) and (1,2) linkages [3,4]. The most common IMOs are isomaltose, panose, isomaltriose, isomaltotetraose and isomaltopentose. As they mostly have α-D-(1,6) linkages in the backbone, IMOs are resistant to the digestive enzymes in the gastrointestinal tract, resulting in low calorific value and a low glycemic index [5]. Furthermore, IMOs are very stable in low pH and moderately high temperature conditions typical for food processing [6].

Functional properties of IMOs on human and animal health are well recognized by many studies. IMOs selectively stimulate the growth or activity of beneficial bacteria or probiotics, in particular *Bifidobacterium* and *Lactobacillus* [7,8], promote peristalsis, prevent and relieve constipation and diarrhea [9] and prevent dental plaque [10]. Due to the stimulation of *Bifidobacterium*, *Lactobacillus* and other probiotics, IMOs indirectly affect various health prevention functions, including anti-tumor function, protection of the liver, improvement of the immune system, reduction in total serum cholesterol and triglycerides and prevention of cardiovascular and cerebrovascular diseases [11,12,13]. Furthermore, the sweetness of commercial IMOs is 60% of the sweetness of sucrose [14]. Due to these physiologically beneficial effects, IMOs are widely used as functional ingredients for food, beverage and pharmaceutical applications.

Further, α-glucosidases are mainly derived from microorganisms, with only a few from plants and animals. Some microorganisms are reported to produce α-glucosidases with transglucosylation activity, including *Aspergillus neoniger* [15], *Aspergillus niger* [16], *Bacillus licheniformis* strain TH4-2 [17], *Geobacillus stearothermophilus* [18], *Xanthophyllomyces dendrorhous* [19] and *Bacillus subtilis* SS-76 [20].

Enzymatic production of oligosaccharides is increasingly used over the alternative chemical method or polysaccharide hydrolysis due to its selectivity, efficiency and environmental friendliness. IMOs are commonly synthesized from maltose and maltooligosaccharides through both hydrolytic and transfer reactions by atypical α-glucosidase (E.C. 3.2.1.20) with transglucosidase activity [3,21,22]. The primary role of this special α-glucosidase with transglucosidase activity is to hydrolyze α-glycosidic linkages from the non-reducing end of maltose and maltoligosaccharides to release α-D-glucose. Thereafter, the enzyme also exhibits transferase activity for the transglucosylation reaction at a higher substrate concentration, resulting in the formation of IMOs. The transfer mainly occurs at the 6-OH group position of the non-reducing end of acceptors; however, it can also occur at the 2-OH, 3-OH and/or 4-OH positions [16]. This transfer to the non-reducing end and the length of oligosaccharides depends on the enzyme specificity and source of α-glucosidase.

Commercially available IMOs are usually a mixture of IMOs with degree of polymerization (DP) ranging from 2 to 6 and lower numbers of α-1,6-glucosidic linkages in the backbone [23]. The limitation of short-chain IMOs such as isomaltose, isomaltotriose, panose and isomaltotetraose is that they can be partially digested and absorbed in the small intestine [24,25]. Moreover, commercial IMOs contain a significant amount of free glucose and/or undesirable sugars [23], which would affect postprandial blood glucose levels [26] and reduce the commercial value of the product. Long-chain IMOs have been reported to be more effective than short-chain IMOs in promoting the growth of probiotics and enhancing health benefits [27,28,29]. Accordingly, many researchers have attempted to search for a highly efficient transferase enzyme with strong transglucosylation activity in order to improve enzymatic properties by protein engineering and/or to use a synergistic enzyme cocktail for producing IMOs with high DP [24,30].

A simple, economical and efficient process is required for industrial-scale production of IMOs. A single-step process using microorganisms as a whole-cell biocatalyst can be a suitable alternative to reduce the cost of enzyme production, recovery–purification and processing time and ultimately boost IMOs production. A few studies have reported the use of microorganisms to produce IMOs from maltose, such as *Microbacterium* sp. [31], *B. subtilis* SS-76 [20], *Debaryomyces hansenii* SCY204 [32] and engineered *Saccharomyces cerevisiae* cells expressing *A. niger* α-glucosidase [33].

*B. subtilis* is preferred for many industrial fermentation processes, especially for food, feed and enzyme production, because it is regarded as non-pathogenic, non-toxicogenic and is approved by the Food and Drug Administration (FDA) as a “generally recognized as safe” (GRAS) organism [34,35]; moreover, it can be easily cultured and controlled to produce extracellular enzymes and fermented products on a large scale. Thus far, there is only one report that demonstrated using *B. subtilis* strain SS-76 for producing short-chain IMOs, such as panose from maltose [20]. Recently, we discovered that *B. sublilis* AP-1 produces high-DP oligosaccharides, possibly IMOs, when grown in maltose medium [36]. In the present research, we studied the production of long-chain IMOs by direct fermentation of maltose using *B. subtilis* strain AP-1. We also evaluated the characteristics, in vitro prebiotic potential, digestibility and cytotoxicity of the IMOs produced.

## 2. Materials and Methods

### 2.1. Materials

Isomaltose, isomaltotriose, maltooligosaccharides (DP2–6), a series of fructooligosacharides (mixed DP2–6), α-amylase (EC 3.2.1.1) from *Aspergillus oryzae* and microbial oligo-α-1,6-glucosidase (EC 3.2.1.10) were purchased from Megazyme Ltd. (Bray, Ireland). De Man, Rogosa and Sharpe (MRS) medium, yeast extract and peptone were obtained from Difco (Detroit, MI, USA). All materials were of standard analytical grade. All other chemicals, media and media components were obtained from Sigma-Aldrich (St. Louis, MO, USA) unless specified otherwise.

### 2.2. Microorganisms, Media and Culture Conditions

*B. subtilis* strain AP-1 was isolated and identified as previously described [30]. The 16s rRNA sequence was deposited in GenBank with accession number MW537819.1. *B. subtilis* strain AP-1 was deposited in Thailand Bioresource Research Center (TBRC), Thailand under accession number TCBR14826. *B. subtilis* strain AP-1 was grown on nutrient agar (NA) at 37 °C under aerobic conditions. To prepare the inoculum, one colony of *B. subtilis* strain AP-1 was picked and transferred to nutrient broth (NB). Then, *B. subtilis* strain AP-1 was cultured at 37 °C and agitated at 200 rpm for 24 h until the optical density (OD) at 600 nm reached around 0.6 (cell concentration of 10^8^ colony-forming units [CFU]/mL). Thereafter, 10% (*v*/*v*) of the inoculum was subcultured on Berg’s mineral salt (MS) fermentation medium supplemented with 50 g/L maltose. The MS medium consists of NaNO_3_ (2 g), K_2_HPO_4_ (0.5 g), MgSO_4_·7H_2_O (0.2 g), MnSO_4_·7H_2_O (0.02 g), FeSO_4_·7H_2_O (0.02 g), CaCl_2_·H_2_O (0.02 g) and 1 L of distilled water [37]. The probiotic bacteria (*Bifidobacterium longum* TISTR 2195 and *Lactobacillus plantarum* TISTR 1465) and pathogenic bacteria (*Escherichia coli* TISTR 117 and *Salmonella enterica* TISTR 1469) were obtained from the Thailand Institute of Scientific and Technological Research (TISTR; Pathumthani, Thailand). *B. longum* TISTR 2195 and *L. plantarum* TISTR 1465 were grown on MRS medium at 37 °C under anaerobic conditions. *E. coli* TISTR 117 and *S. enterica* TISTR 1469 were grown on NB medium at 37 °C under aerobic conditions.

### 2.3. Production of IMOs from Maltose by B. subtilis Strain AP-1

A 10% (*v*/*v*) *B. subtilis* strain AP-1 inoculum was subcultured on 100 mL of MS medium containing 50 g/L maltose. The culture was incubated at 37 °C with agitation at 200 rpm for 72 h. Samples (0.1 mL) were collected at 0, 6, 12, 18, 24, 30, 36, 48 and 72 h, heated at 100 °C for 10 min to inactivate the enzymes and then centrifuged at 10,000× *g* for 10 min. The oligosaccharide products in the culture supernatants were monitored by thin-layer chromatography (TLC). Additionally, the supernatant containing oligosaccharides under optimal condition was fractionated using hollow-fiber ultrafiltration (GE Healthcare Bio-Sciences Corp., Marlborough, MA, USA) with a 10 kDa molecular weight cut-off membrane. Oligosaccharides recovered in the permeate fraction were separated from other compounds of molecular weight higher than 10 kDa. The oligosaccharides were frozen and lyophilized in a laboratory freeze dryer (FD5-T-Series; SIM International Group Co. Ltd., Newark, DE, USA). The dried oligosaccharides were weighted to calculate the yield and used for further analysis. The oligosaccharide yield (%) was defined as the total mass of dried oligosaccharides produced in the culture supernatant based on the mass of maltose used in the fermentation. All fermentation assays were performed in triplicate and samples from the supernatant were analyzed in triplicate.

### 2.4. α-Glucosidase Assay and Protein Determination

The activity of α-glucosidase was measured using maltose as substrate following the methods previously described but with some modifications [38]. The mixture of reaction mixture containing 50 μL of enzyme from *B. subtilis* strain AP-1 and 50 μL of 1% (*w*/*v*) maltose solution in 10 mM phosphate buffer (pH 7.0) was incubated at 40 °C for 15 min. The reaction was heated at 100 °C for 10 min to inactivate enzymes and the concentration of liberated glucose was assessed using a glucose oxidase–peroxidase assay kit (Fujifilm Wako Pure Chemicals, Osaka, Japan). One unit of enzyme activity is defined as the amount of enzyme that produces 1 µmole of glucose per minute under these conditions. The concentration of protein was assessed by the Lowry method [39].

### 2.5. Purification of IMOs Produced from B. subtilis Strain AP-1

The dried oligosaccharides were dissolved in deionized water to a final concentration of 200 mg/mL and centrifuged at 10,000× *g* for 10 min. Samples were separated by size exclusion chromatography (SEC) with a Superdex 30 Increase column (10 mm × 300 mm, GE Healthcare Bio-Sciences AB, Uppsala, Sweden). The column was equilibrated and eluted with deionized water at a flow rate of 0.1 mL/min. Fractions of the elute (0.2 mL) were collected sequentially in test tubes using a fraction collector and monitored by TLC to verify the purity of the oligosaccharides. Those fractions containing pure oligosaccharides were collected and lyophilized for characterization.

### 2.6. TLC

Samples were applied onto aluminum sheet silica gel 60 F254 (Merck, Darmstadt, Germany) and developed using a mixture of *n*-butanol, acetic acid and distilled water in a ratio of 2:1:1. The plates were then treated with a solution containing 1 g of α-diphenylamine dissolved in a mixture of aniline, phosphoric acid and acetone in a ratio of 1:7.5:50. The plates were heated to over 100 °C, leading to appearance of spots [40]. Glucose, maltose, isomaltose and isomaltotriose were used as standards.

### 2.7. Matrix-Assisted Laser Desorption/Ionization Time-of-Flight Mass Spectrometry (MALDI-TOF/MS)

The molecular mass of oligosaccharides was determined by MALDI-TOF/MS on a JMS-S3000 SpiralTOF™ instrument (JEOL, Tokyo, Japan) in positive ion mode. The mass spectrometer was operated at an accelerating voltage of 20 kV. The matrix for oligosaccharide analysis was 2,5-dihydroxybenzoic acid (DHB) in 50% aqueous acetonitrile.

### 2.8. Linkage Analysis by Enzymatic Hydrolysis

The pure oligosaccharide (1 mg/mL) was incubated with excess of each of the hydrolytic enzymes (α-amylase or oligo-α-1,6-glucosidase) for 24 h. Reaction with α-amylase (20 U/mL) was performed in 50 mM sodium acetate buffer at pH 5.0 and 50 °C. Reaction with oligo-α-1,6-glucosidase (20 U/mL) was performed in 50 mM sodium acetate buffer at pH 4.5 and 50 °C. After incubation, the solutions were heated at 100 °C for 10 min to inactivate the enzymes, and the hydrolysis products were determined by TLC. Each treatment was performed in triplicate.

### 2.9. In Vitro Digestibility of IMOs

The in vitro digestibility of IMOs obtained from *B. subtilis* strain AP-1 was tested using the method described by Minekus et al. [41] but with a few changes. The final concentration of stock solution of IMOs was 20 mg/mL, which was obtained by dissolving IMOs in distilled water. Before and after each in vitro digestion treatment, the concentrations of reducing sugar and total sugar were analyzed by the dinitrosalicylic acid (DNS) [42] and the phenol-sulfuric acid methods [43], respectively. The standard curve for determination of sugar content was established using glucose as the internal standard. The calculation of the hydrolysis degree (HD, %) of IMOs in each simulated digestion treatment was performed using Equation (1).
(1)$$\mathrm{HD}\;(\%)=\frac{\mathrm{Rs}\;-\;\mathrm{Ro}}{\mathrm{Ts}\;-\;\mathrm{Ro}}\;\times \;100$$
where R_S_ is the concentration of reducing sugar, T_S_ is the concentration of total sugar after in vitro digestion and R_O_ is the concentration of reducing sugar before treating with the simulated digestion medium.

#### 2.9.1. Simulated Oral Digestion

The simulated oral digestion of IMOs was examined using simulated salivary fluid (SSF, pH 7.0) comprising KCl (15.1 mM), KH_2_PO_4_ (3.7 mM), NaHCO_3_ (13.6 mM), MgCl_2_ (0.15 mM), (NH_4_)_2_CO_3_ (0.06 mM), CaCl_2_ (1.5 mM) and salivary α-amylase [41]. Salivary α-amylase was added to achieve 150 U/mL in the final SSF. Then, 10 mL of IMO stock solution was thoroughly mixed with 10 mL of SSF stock solution and incubated in a water bath at 37 °C for 10 min. During the digestion, 1 mL aliquots of the samples were taken out at 0, 5 and 10 min and heated at 100 °C for 10 min to inactivate the enzymes for further analysis. Each measurement was performed in triplicate and the results were represented as the mean ± SD.

#### 2.9.2. Simulated Gastric Digestion

The simulated gastric fluid (SGF) consisted of KCl (6.9 mM), KH_2_PO_4_ (0.9 mM), NaHCO_3_ (25 mM), MgCl_2_ (0.1 mM), (NH_4_)_2_CO_3_ (0.5 mM), CaCl_2_ (0.15 mM), NaCl (47.2 mM) and gastric pepsin [41]. Gastric pepsin was added to achieve 4000 U/mL in the final SGF. Then, 15 mL of the sample from the simulated oral digestion was mixed with 15 mL of SGF. The mixture was adjusted to pH 3.0 using a small volume of 1.0 M HCl solution and incubated at 37 °C in a water bath for 120 min. After that, 1 mL aliquots of the digested samples were taken out at 0, 30, 60, 90 and 120 h and heated at 100 °C for 10 min to inactivate enzymes for further analysis. Each measurement was performed in triplicate and the results were represented as the mean ± SD.

#### 2.9.3. Simulated Intestinal Digestion

The simulated intestinal digestion of IMOs was carried out using simulated intestinal fluid (SIF) consisting of KCl (6.80 mM), KH_2_PO_4_ (0.80 mM), NaHCO_3_ (85.0 mM), MgCl_2_ (0.33 mM), CaCl_2_ (0.60 mM) and NaCl (38.4 mM) [41]. After the simulated gastric digestion, 20 mL of the digested sample was mixed with 10 mL of SIF solution, 5 mL of a pancreatin solution (based on the trypsin activity at 100 U/mL in the final mixture) and 5 mL of bile salt solution (10 mM). The mixture was adjusted to pH 7.0 using a small volume of 1 M NaOH solution and incubated in a water bath at 37 °C for 4 h. Then, 1 mL aliquots of the digested samples were taken out at 0, 30, 60, 90,120, 180 and 240 min and heated at 100 °C for 10 min to inactivate the enzymes for further analysis. Each measurement was performed in triplicate and the results were represented as the mean ± SD.

### 2.10. In Vitro Fermentation of IMOs by Probiotic and Pathogenic Bacteria

Probiotic bacteria growth (*B. longum* TISTR 2195 and *L. plantarum* TISTR 1465) was conducted in the presence of IMOs, whereas fructooligosaccharides (FOSs) were used as the standard prebiotics. Freshly grown cultures of *B. longum* TISTR 2195 and *L. plantarum* TISTR 1465 were inoculated in MRS medium (pH 6.5) containing 2% (*w*/*v*) IMOs, FOSs or glucose. These cultures were grown at 37 °C for 24 h under anaerobic conditions. The growth of pathogenic bacteria (*E. coli* TISTR 117 and *S. enterica* TISTR 1469) was also conducted in medium containing IMOs. *E. coli* TISTR 117 and *S. enterica* TISTR 1469 were inoculated in NB medium comprising 2% (*w*/*v*) IMOs and grown at 37 °C for 24 h under aerobic conditions. FOSs, which are well-characterized prebiotic oligosaccharides, were selected in this experiment as a positive control, whereas glucose, a non-prebiotic sugar, was used as a non-selective control. The negative control was incubated without sugar. The growth of both probiotic and pathogenic bacteria was performed by spreading the serial dilution of each culture grown on the MRS agar plates for probiotic bacteria and on NA plates for pathogenic bacteria and then incubating at 37 °C for 24 h under anaerobic conditions for probiotic bacteria and aerobic conditions for pathogenic bacteria. The number of colonies was counted as CFU/mL for each culture. The calculation of the prebiotic activity score (PAS) of IMOs and FOSs was performed using Equation (2) [44]:(2)$$\mathrm{PAS}=\frac{{\mathrm{PP}}_{24}-{\;\mathrm{PP}}_{0}}{{\mathrm{PG}}_{24}-{\;\mathrm{PG}}_{0}}-\frac{{\mathrm{EP}}_{24}-{\;\mathrm{EP}}_{0}}{{\mathrm{EG}}_{24}-{\;\mathrm{EG}}_{0}}$$
where PP_24_ and PP_0_ are the number of probiotics (log_10_ CFU/mL) grown on prebiotics for 24 h and 0 h, respectively; PG_24_ and PG_0_ are the number of probiotics (log_10_ CFU/mL) grown on glucose for 24 h and 0 h, respectively; EP_24_ and EP_0_ are the number of *E. coli* TISTR 117 (log_10_ CFU/mL) grown on prebiotics for 24 h and 0 h, respectively and EG_24_ and EG_0_ are the number of *E. coli* TISTR 117 (log_10_ CFU/mL) on glucose for 24 h and 0 h, respectively. Three replicates of each experiment were performed, and the results were presented as mean ± SD.

### 2.11. Short-Chain Fatty Acid (SCFA) Analysis

The grown cultures of probiotic bacteria containing 2% (*w*/*v*) IMOs, FOSs (positive control) and glucose (non-selective control) for 0 h and 24 h were collected and centrifuged at 10,000× *g* for 10 min. The supernatant was filtered through a 0.45 µm membrane filter. Acetic acid, propionic acid and butyric acid contents were analyzed by gas chromatography (GC; Model GC-2014, Shimadzu, Kyoto, Japan) with flame ionization detection. The analytical column was a DB-WAX, size 30 m × 0.32 nm × 0.5 µm (Agilent Technologies, Inc., Santa Clara, CA, USA). Helium gas was used as the carrier gas at a constant flow rate of 3.0 mL/min. Lactic acid content was determined by high-performance liquid chromatography (HPLC; Model LC-20A, Shimadzu, Kyoto, Japan) and using an Aminex HPX-87H column (Bio-Rad, Hercules, CA, USA) and an SPD-M20A diode array detector (Shimadzu, Kyoto, Japan). The mobile phase of 5.0 mM sulfuric acid was passed through the column at a flow rate of 0.6 mL/min, keeping the oven temperature at 50 °C. The wavelength for lactic acid detection was set at 210 nm. Three replicates of each experiment were performed, and the results were presented as mean ± SD.

### 2.12. In Vitro Cytotoxicity Assay of IMOs on Mammalian Cells

The in vitro cytotoxic effect of the IMOs obtained from *B. subtilis* strain AP-1 on mouse fibroblast (L929) cells was assessed using the 3-(4,5-dimethylthiazol-2-yl)-2,5-diphenyl tetrazolium bromide (MTT) assay [45] but with some modification. The viability of cells was determined by the reduction of MTT by mitochondrial dehydrogenases to the water-insoluble pink formazan compound. Cells were cultured in Dulbecco’s Modified Eagle’s Medium (DMEM) (Gibco Laboratories, Grand Island, NY, USA) supplemented with 10% fetal bovine serum (FBS). Then, 100 µL of cells (1.0 × 10^4^ cells/mL) were seeded in 96-well plates and incubated at 37 °C in 5% CO_2_ for 24 h. After 24 h of incubation, the cells were treated with 100 μL of IMOs at concentrations of 0.31, 0.62, 1.25, 2.5, 5 and 10 mg/mL. The plates were incubated at 37 °C in 5% CO_2_ for 24 h. After incubation, 20 µL of MTT solution (5 mg/mL) were added to each well, and the plates were incubated for an additional 4 h. The medium was subsequently removed, and 100 μL of dimethyl sulfoxide (DMSO) was added to dissolve the formazan crystals. The absorbance was measured at 560 nm and the percentage of cell viability was calculated from Equation (3):(3)$$\%\;\mathrm{Cell}\;\mathrm{viability}=\frac{{\mathrm{Absorbance}}_{\mathrm{sample}}}{{\mathrm{Absorbance}}_{\mathrm{control}}}\;\times \;100$$
where Absorbance_control_ is the absorbance of cells treated with 1% DMSO and Absorbance_sample_ is the absorbance of cells treated with samples. Each measurement was performed in triplicate and the results were represented as the mean ± SD.

### 2.13. Statistical Analysis

Data were expressed as the mean ± SD in triplicate. The results were analyzed using IBM SPSS Statistics (SPSS Inc., Chicago, IL, USA). Statistical significance was determined by two-way ANOVA followed by the Tukey test, with *p* < 0.05 considered to be statistically significant.

## 3. Results and Discussion

### 3.1. Production of IMOs by B. subtilis Strain AP-1

To investigate the time course of IMOs production, we cultivated the *B. subtilis* strain AP-1 in MS medium containing 50 g/L maltose. Maltose has been reported as an excellent acceptor for the production of IMOs by α-glucosidases with transglucosidase activity [31,46]; therefore, maltose was chosen as a carbon source in this study. As shown in Figure 1, at 6 h of cultivation, most of the maltose was hydrolyzed to glucose and oligosaccharides of up to DP4 were produced. We found that, at 6 h, oligosaccharide products (DP2 and DP3) with the same magnitude as IMO2 and IMO3 were detected; therefore, these products should be the series of IMOs. Between 6 and 18 h, the amount of maltose continued to decrease and glucose and oligosaccharides with higher DP were detected. At 24 h, maltose was completely hydrolyzed and the length of oligosaccharides increased up to DP8 (Figure 1). The results indicated the presence of α-glucosidase activity together with transglucosidase activity produced by *B. subtilis* strain AP-1.

The activity of α-glucosidase was detected at 1.273 U/mL or 0.533 U/mg protein in the culture supernatant. Further, α-glucosidase catalyzed the release of glucose from maltose and then catalyzed transglucosylation, which transferred the glucosyl moiety from the glucose enzyme intermediate to the OH group of acceptors, such as glucose, maltose and oligosaccharides, present in the culture, resulting in the formation of disaccharides, trisaccharides and longer oligosaccharides, respectively [3,21]. These oligosaccharides could be used as acceptors and then further transglucosylated to produce oligosaccharides of higher chain length. Figure 1 shows that oligosaccharides of higher DP were accumulated in the culture medium after a long cultivation time. In contrast, glucose, maltose and short-chain oligosaccharides were continuously reduced. Furthermore, it was noteworthy that maltose and glucose were depleted at 36 h of cultivation and the length of oligosaccharides increased up to more than DP10. The pattern of oligosaccharides in the culture supernatant did not change after 36 h of cultivation, possibly due to glucose deficiency. This phenomenon is advantageous for the production of IMOs because of their high purity without glucose and maltose contamination. In contrast to other studies on the direct use of microorganisms to produce IMOs, maltose and glucose were found to remain in the final product. Ojha et al. [31] demonstrated the use of cell-bound α-glucosidase of *Microbacterium* sp. with 10 IU of α-glucosidase activity for production of IMOs from 10% (*w*/*v*) maltose. The IMO product contained isomaltotriose, isomaltotetraose, isomaltopentaose, isomaltohexaose and about 50% remaining maltose. Rengarajan and Palanivel [32] isolated the yeast strain *D. hansenii* SCY204, which efficiently utilized maltose and converted it into IMOs. It able to produce 4.02 U/mL of α-glucosidase activity and IMOs with DP2–6 in culture medium containing 30% (*w*/*v*) maltose; however, the remaining maltose and glucose were also accumulated in the culture. Furthermore, Casa-Villegas et al. [33] developed engineered *S. cerevisiae* cells expressing *A. niger* α-glucosidase to produce IMOs for a one-step process. In culture medium containing 30% (*w*/*v*) maltose, it can produce 1.4 mU/mL of α-glucosidase activity and short-chain IMOs, namely panose and isomaltose, as major products; however, a high amount of glucose remained in the culture. Additionally, common approaches for the production of IMOs by enzymatic method often result in the accumulation of undesirable sugars, particularly glucose and maltose, in the final product [23,47]. Health benefits and commercial value of product can be significantly reduced by these digestible sugars.

The removal of glucose or undesirable sugars from IMOs products can be accomplished by chromatography [48] and nanofiltration [49], but these methods are expensive and have low efficiency. By comparison, yeast fermentation is an inexpensive and widely used method of purification. For instance, immobilized cells of *Zymomonas mobilis* were used to eliminate glucose, fructose and sucrose from IMO mixtures [50]. *S. cerevisiae* was used to remove glucose, while *Saccharomyces carlbergensis* was used to remove glucose, maltose and maltotriose from IMO mixtures produced either from rice crumbs or tapioca flour [47]. Furthermore, glucose contamination in IMOs product obtained from the yeast *D. hansenii* SCY204 was removed by *S. cerevisiae* in a second step [32]. However, this additional yeast fermentation step increases the manufacturing costs of IMOs, and some metabolites generated by yeast during fermentation may affect the quality and purity of the IMOs products. Therefore, *B. subtilis* strain AP-1 could be an effective strain for long-chain IMO production with less unwanted sugar contamination, especially glucose and maltose.

As TLC is limited to the detection of long-chain oligosaccharides, the molecular mass and DP of the oligosaccharides obtained from culture supernatant at 36 h of cultivation (free of glucose and maltose) were confirmed by MALDI-TOF/MS. 

The results demonstrated the presence of different oligosaccharides with pseudo-molecular ion peaks at *m*/*z* values of 365.1, 527.2, 689.3, 851.3, 1013.4, 1176.5, 1338.6, 1500.6, 1662.4, 1824.3, 1986.4, 2148.4 and 2311.0 (including one sodium adduct) and 381.3, 543.2, 705.2, 867.3, 1029.4, 1192.4, 1354.5, 1516.4, 1678.2, 1840.1, 2002.1, 2164.5 and 2327.0 (including one potassium adduct), corresponding to oligomers of glucose units with DP ranging from 2 to 14, as shown in Figure 2. The intensity of the IMOs was in the order DP7 > DP6 > DP8 > DP5 > DP9 > DP4 > DP10 > DP3 > DP2 > DP11 > DP12 > DP13 > DP14. The total yield of the IMOs without contamination from glucose and maltose was 36.33 g/L or 72.66%. Here, *B. subtilis* strain AP-1 produced a series of long-chain IMOs with up to DP14 from maltose through its α-glucosidase, its strong hydrolytic activity toward maltose and its high transglucosylation activity enabling the synthesis of longer IMO chains.

### 3.2. Purification and Linkage Analysis of IMOs

Although the transglucosylation action of α-glucosidases mainly occurs at position 6-OH of the non-reducing end of acceptors, in some cases, it can occur at positions 2-OH, 3-OH and/or 4-OH [16]. To confirm that the IMOs obtained from *B. subtilis* strain AP-1 consisted of α-1,6-glycosidic linkages, each IMO was purified by SEC to analyze its structure. The IMO mixture from the culture supernatant of the strain AP-1 was subjected to SEC with a Superdex 30 Increase column and eluted with deionized water at a flow rate of 0.1 mL/min. The eluent was collected in 100 fractions and the IMO elution profile was detected on TLC. As IMOs with DP higher than 10 had low concentrations, this study isolated and identified IMOs in the DP range 2–10. The IMO products were observed on TLC after fraction number 50 and IMOs with DP10, 9, 8 and 2 were separated into fraction numbers 51–55, 57–59, 61–64 and 88–90, respectively, as shown in Figure 3A. The IMO mixtures with DP2–7 and DP2–4 present in fraction numbers 73–82 and 83–84 were pooled and concentrated by lyophilization and named F1 and F2, respectively. The F1 and F2 samples were reloaded on the Superdex 30 Increase column for SEC, decreasing the flow rate to 0.025 mL/min to improve separation and resolution, and the eluent of samples F1 and F2 was collected. TLC of the F1 elution profile showed that IMOs with DP7, 6, 5, 3 and 2 were isolated in fraction numbers 250–256, 260–263, 271–274, 296–299 and 310–316, respectively (Figure 3B), whereas TLC of the F2 elution profile showed that IMOs with DP4, 3 and 2 were isolated in fraction numbers 267–276, 281–287 and 295–313, respectively (Figure 3C). Finally, single IMOs from IMOs of size DP2–10 were isolated and investigated further for the linkages of oligosaccharides. 

The molecular mass of each purified IMO was confirmed by MALDI-TOF/MS. The results showed the pseudo-molecular ion peak at *m*/*z* values of the nine purified IMOs, corresponding to IMOs with DP2–10 (Figure 4). Then, the linkages of the purified IMOs were investigated by specific enzymatic action and the degradation products were analyzed by TLC. Further, α-amylase and oligo-α-1,6-glucosidase were chosen to digest these oligosaccharides; α-amylase randomly hydrolyzes the internal α-1,4-glucosidic linkages of starch and oligomers of glucose, whereas oligo-α-1,6-glucosidase catalyzes the degradation of α-1,6-glycosidic linkages in oligosaccharides. When both enzymes were tested with maltohexaose (M6), the results showed that α-amylase hydrolyzed maltohexaose and then released maltose and maltotriose as products, whereas oligo-α-1,6-glucosidase could not degrade maltohexaose, indicating the presence of α-1,4-glycosidic linkages within the maltohexaose (Figure 5A). In contrast, all nine purified IMOs (DP2–10) obtained from *B. subtilis* strain AP-1 were strongly resistant to the endo α-1,4-hydrolyzing activity of α-amylase, whereas they were completely catalyzed by oligo-α-1,6-glucosidase and the product contained only glucose (Figure 5B). Therefore, the series of oligosaccharides obtained from *B. subtilis* strain AP-1 should be IMOs consisting only of α-1,6-glycosidic linkages. These results implied that the maltose-medium-grown *B. subtilis* strain AP-1 produced α-glucosidase, which strongly hydrolyzed the α-1,4-glycosidic linkage of maltose to release glucose. Subsequent transglucosylation of glucose moieties to the non-reducing end of glucose residues [3] via the α-1,6-glycosidic linkage led to isomaltose production. Elongation of the oligosaccharide chains by transglucosylation generated long linear IMOs with α-1,6-glycosidic linkages. As the α-(1,6) glycosidic linkages of IMOs are strongly resistant to human gastrointestinal tract digestion and are retained in the colon to promote beneficial probiotic bacteria [5,30], the IMO products obtained from *B. subtilis* strain AP-1 probably have good prebiotic properties. Most commercial IMO products contain IMOs with DP ranging from 2 to 6 and with lower numbers of α-1,6-glucosidic linkages [23], along with a significant amount of free glucose that can increase the blood sugar level [25]. Therefore, this research proposed the production of a series of IMOs up to DP14 using *B. subtilis* strain AP-1 in maltose-containing medium. Moreover, these IMOs consisted of α-1,6-glucosidic linkages without contamination from glucose and maltose, unlike the commercially available IMOs. 

### 3.3. In Vitro Determination of Prebiotic Properties of IMOs

#### 3.3.1. In Vitro Digestibility of IMOs

The resistance to upper gastrointestinal tract digestion is considered as one of the crucial properties of prebiotic compounds. Therefore, the IMOs produced from *B. subtilis* strain AP-1 were first evaluated for in vitro digestibility using the standardized in vitro method [41]. This method was also applied to evaluate the digestibility of several prebiotic carbohydrates, non-starch polysaccharides and oligosaccharides [44,51].

For the oral phase, salivary α-amylase is the first digestive enzyme in contact with foods, which can break down starch or other carbohydrates containing α-(1,4)-glycosidic bonds and promote the digestion and absorption of foods. The initial reducing sugar content of IMO solution from *B. subtilis* strain AP-1 was 1.40 mg/mL. The results showed no significant changes in the reducing sugar content or HD (%) after digestion for 10 min at pH 7.0 and 37 °C (Table 1). The results indicated that IMOs were resistant to α-amylase. The presence of only the α-(1,6)-glycosidic bonds in IMOs makes them resistant to cleavage by α-amylase, which only cleaves α-(1,4)-glycosidic linkages.

For the gastric phase, the IMOs solution from digestion in the oral phase was sequentially subjected to digestion by simulated gastric fluid under acidic conditions at pH 3.0. As shown in Table 1, the reducing sugar content of the IMOs solution before digestion was 0.71 mg/mL. The reducing sugar content and HD did not change after digestion for 120 min at 37 °C. The IMOs resistance to hydrolysis at low pH of the gastric juice indicates their availability in the small intestine of the gastrointestinal tract. These results are in accordance with other enzyme-derived IMOs, which showed high stability under acidic conditions at pH 3.0 [6,24].

Prebiotics must resist the bile juice and digestive enzymes of the intestine and reach the probiotic bacteria present in the colon intact. Therefore, the IMOs solution from the oral and stomach phases was sequentially reacted with simulated intestinal fluid at pH 7.0. The initial reducing sugar content of the IMOs solution was 0.34 mg/mL. The results showed that the reducing sugar content and HD slightly increased with time in the intestinal fluid: maximum reducing sugar content was 0.38 mg/mL and maximum HD was 2.45%, found at 240 min of incubation in the intestinal fluid at 37 °C (Table 1). This result indicated that the IMOs could tolerate the intestinal fluid of the small intestine, with over 97.55% still available intact as a carbon source for probiotic bacteria. However, partial digestion of IMOs may occur when short-chain IMOs, such as isomaltose and isomaltotriose, are hydrolyzed by amylolytic enzymes in pancreatin. Several studies have reported the hydrolysis of short-chain IMOs such as isomaltose and isomaltotriose by digestive enzymes in the intestinal epithelium while the long-chain IMOs remained stable [5,24]. The IMOs obtained from *B. subtilis* strain AP-1 could pass through the digestive system and reach the probiotic bacteria present in the colon intact.

Long-chain IMOs with DP of 7 or higher have better prebiotic and health beneficial effects than short-chain IMOs because they are resistant to upper gastrointestinal tract digestion and are, therefore, retained in the colon to stimulate beneficial probiotic bacteria [27,52]. Moreover, long-chain IMOs have been demonstrated to promote quercetin-3-glucoside in the lumen [53] and to enhance the synthesis of tumor necrosis factor-alpha in primary macrophages through toll-like receptor 4 signaling with pro-inflammatory effects [28]. Thus far, the maximum DP of IMOs produced using microorganisms, namely *Microbacterium* sp. [31] and the yeast strain *D. hansenii* SCY204 [32], has been reported to be DP6. Therefore, this is the first report on the production of a series of IMOs with DP up to 14 from maltose using *B. subtilis* strain AP-1. It is a simple and economical method of producing a series of long-chain IMOs without adding any external enzymes.

#### 3.3.2. Effect of IMOs on the Growth of Probiotic and Pathogenic Bacteria 

Upon reaching the colon, prebiotics help to stimulate the growth of beneficial or probiotic bacteria, such as *Bifidobacteria* and *Lactobacillus*. We investigated the growth of probiotic bacteria *B. longum* TISTR 2195 and *L. plantarum* TISTR 1465 in 2% (*w*/*v*) IMOs at 37 °C for 24 h. FOSs, which are well-known prebiotic oligosaccharides, were chosen as the positive control, whereas glucose, a non-prebiotic sugar, was used as the non-selective control. The growth of probiotic bacteria in the culture containing IMOs was comparable to that of the culture containing FOSs and glucose (Table 2). IMOs stimulated the growth of *B. longum* TISTR 2195 and *L. plantarum* TISTR 1465, resulting in a significant increase in the number of cells from 7.79 to 9.41 and from 7.86 to 9.28 log_10_ CFU/mL, respectively, in 24 h. In contrast, FOSs and IMOs did not promote the growth of pathogenic bacteria *E. coli* TISTR 117 and *S. enterica* TISTR 746 (Table 2). These results demonstrated that IMOs could stimulate the growth of probiotic bacteria but not pathogenic bacteria, exhibiting typical characteristics of prebiotic compounds.

Probiotic bacteria grow well in the culture containing prebiotics because they can produce the required enzymes to digest prebiotic oligosaccharides [54]. On the other hand, pathogenic bacteria *E. coli* and *S. enterica* lack the necessary enzymes to digest prebiotic oligosaccharides, which makes them unable to utilize prebiotics for growth. The prebiotic activity score or PAS is defined as the ability of a given compound to promote the growth of probiotics relative to non-probiotics and non-prebiotic substrates such as glucose [44]. The PAS of IMOs and FOSs were calculated based on Equation (2): IMOs showed PAS of 0.63 and 0.56 for *B. longum* TISTR 2195 and *L. plantarum* TISTR 1465, respectively, and FOSs displayed PAS of 0.45 and 0.74 for *B. longum* TISTR 2195 and *L. plantarum* TISTR 1465, respectively. IMOs were more favorably utilized by *B. longum* than FOSs, probably because *B. longum* produced highly efficient enzymes for their degradation. Furthermore, IMOs obtained from *B. subtilis* strain AP-1 had DP up to 14, while FOSs had DP up to 6. It has been reported that *Bifidobacteria* preferentially utilize oligosaccharides with higher DP, whereas *Lactobacillus* sp. preferentially metabolize short-chain oligosaccharides [55]. These results showed that IMOs from *B. subtilis* strain AP-1 stimulate the growth of probiotic bacteria, in particular *Bifidobacteria*. Thus, IMOs can be used as prebiotic compounds to improve human intestinal health.

#### 3.3.3. Production of SCFAs by Probiotic Bacteria

SCFAs are the main metabolites produced by gut microbiota during fermentation of partially and non-digestible polysaccharides and/or oligosaccharides. They have a significant effect on the balance of gut microbiota growth and functions, thereby contributing to the health of the host [56]. In this study, the production of SCFAs by *B. longum* TISTR 2195 and *L. plantarum* TISTR 1465 cultures containing IMOs obtained from *B. subtilis* strain AP-1 were analyzed at 0 h (baseline) and 24 h of incubation and compared with the control (without a carbon source) and FOSs. The total SCFA concentrations of *B. longum* and *L. plantarum* cultures with IMOs and FOSs markedly increased at 24 h and were higher than the controls, indicating that both IMOs and FOSs had significant promoting effects on SCFA production (Table 3).

The *B. longum* culture containing IMOs (10.62 mM) from *B. subtilis* strain AP-1 showed higher total SCFA content than with FOSs (6.92 mM), whereas the production of total SCFAs by *L. plantarum* with FOSs (14.88 mM) was higher than with IMOs (12.96 mM) (Table 3). These results may correlate with changes in the number of probiotic bacteria in the cultures because *B. longum* grew better on IMOs (9.41 log_10_ CFU/mL) than on FOSs (8.98 log_10_ CFU/mL), whereas *L. plantarum* thrived better on FOSs (9.77 log_10_ CFU/mL) than on IMOs (9.28 log_10_ CFU/mL) (Table 2). Furthermore, both probiotic bacteria produced lactic, acetic, propionic and butyric acids as metabolites during the fermentation. Acetic acid was the predominant SCFA produced from *B. longum,* whereas lactic acid was the major SCFA from *L. plantarum* (Table 3). However, it was notable that IMOs promoted the production of acetic, propionic and butyric acids by both probiotic bacteria better than FOSs.

SCFAs have been demonstrated to play a crucial role in maintaining good health and preventing disease. Lactic acid can modulate the immune system and inhibit histone deacetylases [57]. Lactic acid and acetic acid can be utilized further by cross-feeding bacteria in the colon to produce butyric acid [58]. A high concentration of acetic acid may prevent the synthesis of fat, which may contribute to obesity [59]. Propionic acid can be absorbed in the colon and further positively influence liver function and cholesterol metabolism [60]. Butyric acid can serve as an energy source for colonic epithelium [61]. Furthermore, it can control growth and apoptosis of both epithelial and immune cells [62] and inhibit colitis and colonic cancers [63,64]. Overall, these results demonstrated that the series of long-chain IMOs obtained from *B. subtilis* strain AP-1 can promote the growth of beneficial probiotic bacteria and enhance the production of SCFAs, such as acetic acid, lactic acid, propionic acid and butyric acid.

### 3.4. In Vitro Cytotoxicity of IMOs with Mammalian Cells

An in vitro cytotoxicity assay was performed to evaluate whether the IMOs obtained from *B. subtilis* strain AP-1 were safe for mammalian cells (L-929), and then a comparison to commercial prebiotic FOSs was determined. The samples were prepared using a two-fold serial dilution method at concentrations of 10, 5, 2.5, 0.625 and 0.313 mg/mL. The viability of L-929 cells was assessed by the MTT assay. The IMOs and FOSs showed no effect on the survival of L-929 cells at concentrations up to 5 mg/mL and both were comparable up to 5 mg/mL (Figure 6). At 10 mg/mL, IMOs and FOSs showed decreased survival of 75% and 25%, respectively. At less than 5 mg/mL, the IMOs obtained from *B. subtilis* strain AP-1 proved to be non-toxic to mammalian cells.

## 4. Conclusions

This study proposes the production of a series of long-chain IMOs in which glucose residues are connected with α-1,6-glycosidic linkages up to DP14, without contamination from glucose and maltose by direct fermentation of maltose using *B. subtilis* strain AP-1. These IMOs exhibit good prebiotic properties, such as tolerance to acidic conditions and digestive enzymes of the gastrointestinal tract, stimulation of probiotic bacteria growth to produce short-chain fatty acids and no stimulating effect on pathogenic bacteria growth. Our results have demonstrated a simple, economical and promising method of producing long-chain IMOs with prebiotic potential. 

## Figures and Tables

**Figure 1 foods-12-01499-f001:**
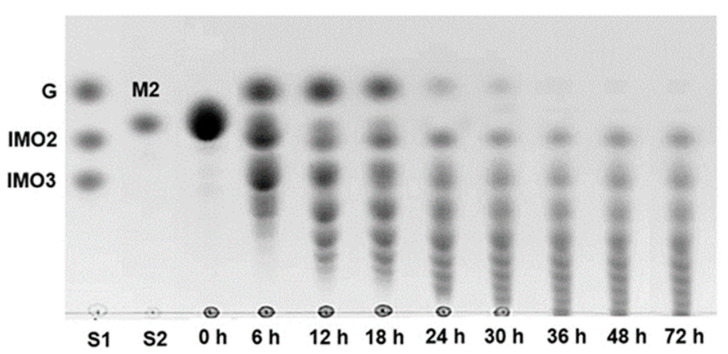
TLC patterns of culture supernatant of *B. subtilis* strain AP-1 grown in MS medium containing 5% (*w*/*v*) maltose. Lane S1: glucose (G), isomaltose (IMO2) and isomaltotriose (IMO3) standard; lane S2: maltose (M2) standard.

**Figure 2 foods-12-01499-f002:**
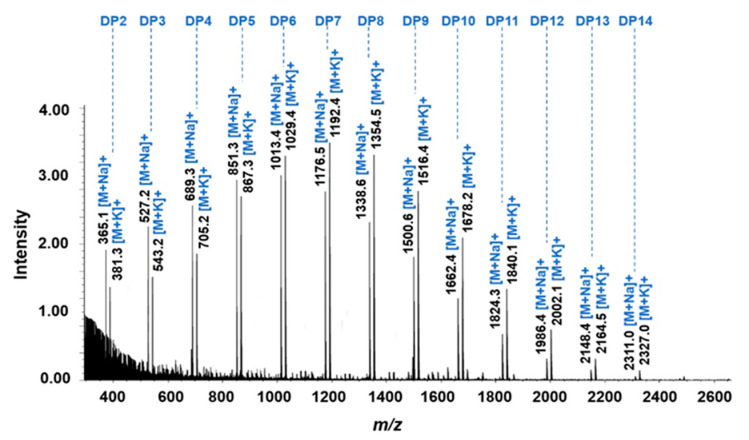
MALDI-TOF-MS spectra of oligosaccharides obtained from the culture supernatant of *B. subtilis* strain AP-1 at 36 h.

**Figure 3 foods-12-01499-f003:**
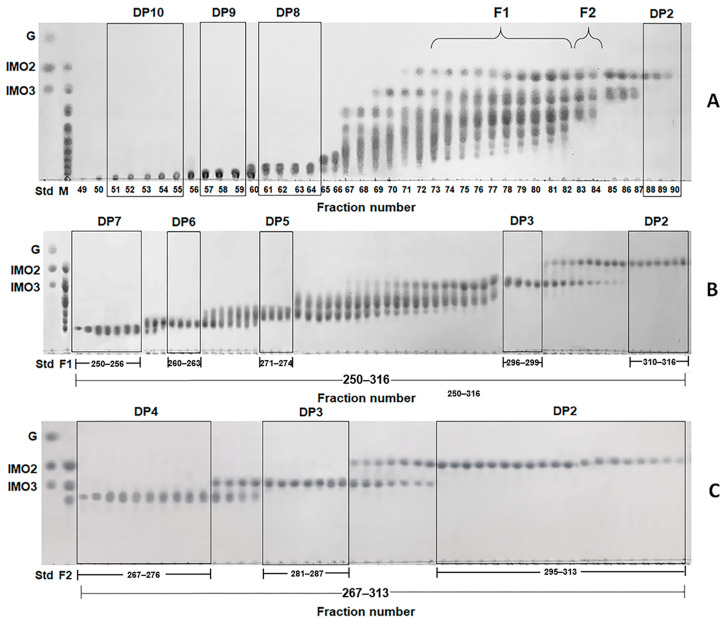
(**A**) TLC patterns of elution profile of IMOs obtained from the culture supernatant of *B. subtilis* strain AP-1 at 36 h using a Superdex 30 Increase column with a flow rate of 0.1 mL/min. (**B**,**C**) TLC patterns of elution profiles of samples F1 and F2, respectively, using a Superdex 30 Increase column with a flow rate of 0.025 mL/min. Lane Std: glucose (G), isomaltose (IMO2) and isomaltotriose (IMO3) standard; lane M: IMO mixture from culture supernatant at 36 h of *B. subtilis* strain AP-1; lane F1: sample F1; lane F2: sample F2.

**Figure 4 foods-12-01499-f004:**
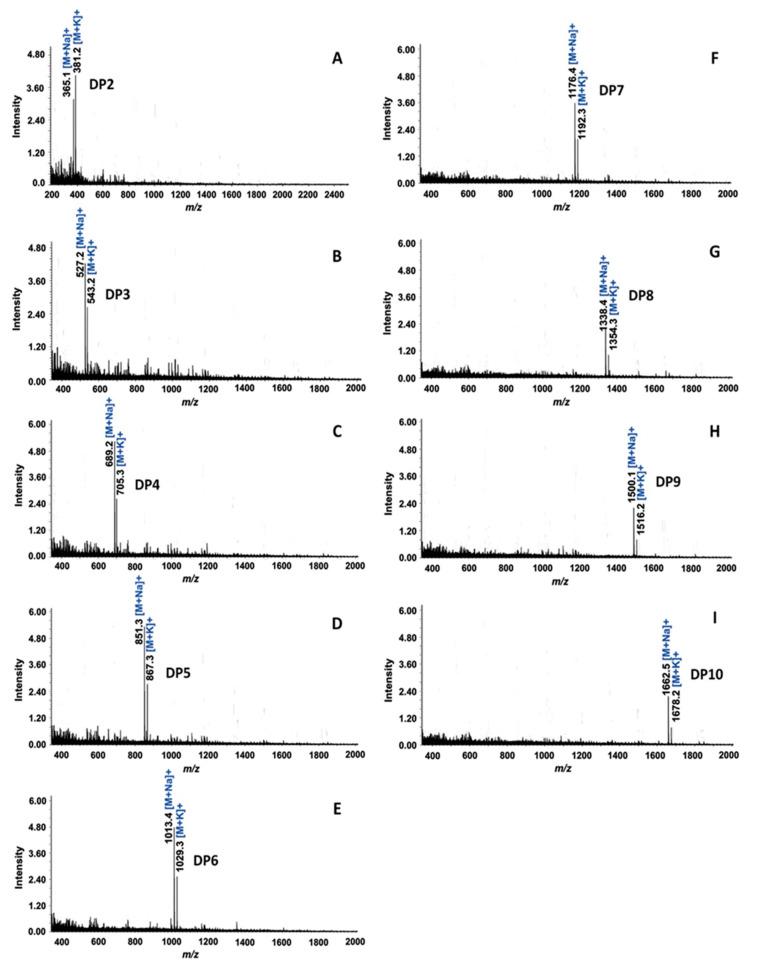
MALDI-TOF-MS spectra of purified IMOs produced from *B. subtilis* strain AP-1 with degree of polymerization (DP) as follows: (**A**) DP2, (**B**) DP3, (**C**) DP4, (**D**) DP5, (**E**) DP6, (**F**) DP7, (**G**) DP8, (**H**) DP9 and (**I**) DP10.

**Figure 5 foods-12-01499-f005:**
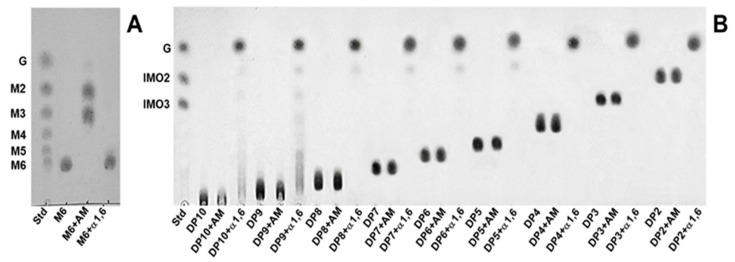
TLC of enzymatic hydrolysis of maltohexaose (M6) (**A**) and each IMO with degree of polymerization (DP) 2–10 produced from *B. subtilis* strain AP-1 (**B**) by the action of α-amylase (AM) and oligo-α-1,6-glucosidase (α1,6). Glucose (G), maltooligosaccharides (M2–M6), isomaltose (IMO2) and isomaltotriose (IMO3) were used as standards (shown on the left-hand side of each image).

**Figure 6 foods-12-01499-f006:**
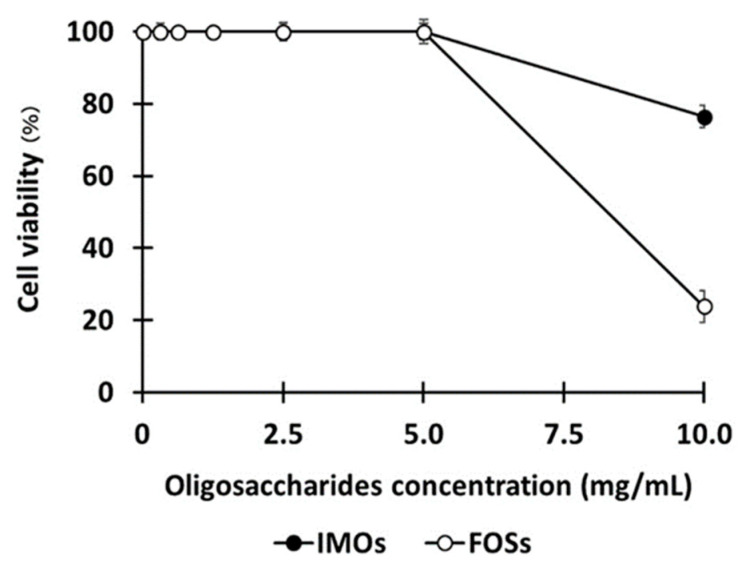
Effect of oligosaccharides on the cell viability of mammalian cells. Data are the means of three independent experiments and error bars represent the standard deviation. IMOs, isomaltooligosaccharides; FOSs, fructooligosaccharides.

**Table 1 foods-12-01499-t001:** Reducing sugar content and hydrolysis degree of IMOs during the oral, gastric and intestinal phases of digestion.

Treatments	Time (min)	Reducing Sugar Content (mg/mL)	Hydrolysis Degree (HD, %)
Oral phase	0	1.40 ± 0.02 ^a^	NA
	5	1.42 ± 0.06 ^a^	0.22 ± 0.02 ^a^
	10	1.42 ± 0.08 ^a^	0.21 ± 0.05 ^a^
Gastric phase	0	0.71 ± 0.01 ^a^	NA
	30	0.71 ± 0.02 ^a^	0.26 ± 0.07 ^a^
Intestinal phase	0	0.34 ± 0.01 ^a^	NA
	30	0.36 ± 0.05 ^ab^	1.06 ± 0.23 ^a^
	60	0.37 ± 0.03 ^ab^	1.91 ± 0.10 ^ab^
	90	0.38 ± 0.06 ^b^	2.21 ± 0.17 ^b^
	120	0.38 ± 0.02 ^b^	2.38 ± 0.21 ^b^
	180	0.38 ± 0.04 ^b^	2.45 ± 0.28 ^b^
	240	0.38 ± 0.11 ^b^	2.45 ± 0.35 ^b^

Different letters indicate significant differences among different times (Tukey test, *p* < 0.05) in the same digestive phase. NA = not available.

**Table 2 foods-12-01499-t002:** Growth of probiotic and pathogenic bacteria in the medium without a carbon source and with glucose, FOSs and IMOs (number of cells reported as log_10_ CFU/mL).

Bacterium	0 h	24 h			
		No Sugar	2% (*w*/*v*) Glucose	2% (*w*/*v*) FOSs	2% (*w*/*v*) IMOs
*B. longum*	7.79 ± 0.02 ^a^	7.81 ± 0.03 ^a^	10.34 ± 0.05 ^d^	8.98 ± 0.03 ^b^	9.41 ± 0.03 ^c^
*L. plantarum*	7.86 ± 0.02 ^a^	7.91 ± 0.01 ^a^	10.38 ± 0.04 ^d^	9.77 ± 0.02 ^c^	9.28 ± 0.03 ^b^
*E. coli*	7.61 ± 0.02 ^a^	7.62 ± 0.08 ^a^	9.51 ± 0.03 ^b^	7.62 ± 0.05 ^a^	7.58 ± 0.03 ^a^
*S. enterica*	7.44 ± 0.04 ^a^	7.50 ± 0.04 ^a^	9.79 ± 0.08 ^b^	7.42 ± 0.31 ^a^	7.39 ± 0.01 ^a^

Different letters indicate significant differences (Tukey test, *p* < 0.05) in the same row.

**Table 3 foods-12-01499-t003:** Production of SCFAs by *B. longum* and *L. plantarum* grown at 37 °C for 24 h. FOSs, fructooligosaccharides; IMOs, isomaltooligosaccharides obtained from *B. subtilis* strain AP-1.

Bacterium	SCFAs (mM)	No Sugar	2% (*w*/*v*) FOSs	2% (*w*/*v*) IMOs
*B. longum*	Lactic acid	0.15 ± 0.01 ^a^	2.63 ± 0.02 ^c^	2.56 ± 0.01 ^b^
	Acetic acid	0.07 ± 0.01 ^a^	3.53 ± 0.18 ^b^	6.55 ± 0.21 ^c^
	Propionic acid	0.14 ± 0.04 ^a^	0.55 ± 0.04 ^b^	1.11 ± 0.08 ^c^
	Butyric acid	0.06 ± 0.01 ^a^	0.21 ± 0.02 ^b^	0.40 ± 0.03 ^c^
*L. plantarum*	Lactic acid	0.20 ± 0.01 ^a^	13.06 ± 0.02 ^c^	9.09 ± 0.02 ^b^
	Acetic acid	0.07 ± 0.01 ^a^	1.13 ± 0.07 ^b^	2.65 ± 0.05 ^c^
	Propionic acid	0.15 ± 0.01 ^a^	0.55 ± 0.03 ^b^	0.89 ± 0.03 ^c^
	Butyric acid	0.06 ± 0.01 ^a^	0.14 ± 0.02 ^b^	0.33 ± 0.02 ^c^

Different letters indicate significant differences (Tukey test, *p* < 0.05) in the same row.

## Data Availability

Data are contained within the article.

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
