# Peer review of "Production of a Series of Long-Chain Isomaltooligosaccharides from Maltose by Bacillus subtilis AP-1 and Associated Prebiotic Properties"

_foods, 2023, doi:10.3390/foods12071499_

Round 1

Reviewer 1 Report

In overall, I think it is a well written paper, however, there are some methodological errors that should be discussed in the revised version of the paper. Introduction is written thoroughly and proper language is used in the manuscript.

In lines 35-36 authors state that IMOS are the most promising prebiotics while actually inulin is considered to be a golden standard. Therefore, I would recommend to clarify, that authors meant novel prebiotics.

Line 132: what was the total volume of the starting culture and and what was the volume of collected samples? Reducing the volume significantly could completely change growth conditions due to changing gas transfer. Those details should be added.

Line 141: what was the method used for dry weight determination? That should be added.

Line 171: Why only those standards were used? They do not represent all IMOs.

Line 210: What was the volume of 1 M HCl? In case if it was significant, it could have changed significantly the volume of the reaction mixture.

Lines 219-220: It is best to prepare pancreatic solution or bile solution with the buffer, otherwise adjusting the pH afterwards can dilute the whole reaction mixture too much. It should be included in the discussion that the method of the in vitro digestion should be improved.

Lines 229-231: In the case of growth studies it’s best to choose minimum nitrogen base medium and add IMOs as the only carbon source. In that way it would be certain that those compounds are truly digestible by tested strains. Moreover, if minimum nitrogen base was used, it would be clearer what quantities of FAS could be produced from IMOs.

Lines 267-268: Why fribroblasts were used? Since those prebiotics could be potentially used for food or cosmetics, it is more justified to use the cells of colon or skin epithelium.

Line 286: The use of Tukey test is very much justified but it requires 5 replicates. Why experiments were carried out only in triplicates?

Lines 300-301: I don’t think that there is enough evidence to state that the enzyme had transglucosidase activity since a very limited number of standards was used.

Lines 310-320: I don’t think that it is justified to state that long chain oligosaccharides was present since the number of standards for TLC was very limited.

Lines 321-333: Names of microorganisms should be written in italics.

Lines 560-561: IMOs were tested only on one cell line so it is not justified to say in general that they are safe for human cells.

Reviewer 2 Report

Comments for authors

The paper entitled “Single-Step Production of a Series of Long-Chain 2 Isomaltooligosaccharides from Maltose by Bacillus subtilis 3 AP-1 and Associated Prebiotic Properties” is very interesting research; however, it is necessary to adjust throughout the text:

1.       In section 2.3 indicated that the inoculum was adjusted at 10% (v/v), which is necessary to indicate cell concentration (CFU/mL) for the reproducible experiment. The introduction should be adjusted to the title since a selective determination is indicated and is not contextualized in the introduction.

2.       Section 2.4 is necessary to describe the reaction system and when indicate the reaction mixture that contained 50 μL of enzyme and 50 μL, what was de enzyme origin (commercial or enzymatic extract?).

3.       Result and discussion section. Maltose has been reported as an excellent acceptor for the production of IMOs by α- glucosidases with transglucosidase activity, therefore maltose was chosen as a carbon source in this study, what is the context of this paragraph, it is very confusing, and it is the beginning of the results, what do you intend with this wording?

4.       line 302 to 308, it is meaningless to describe the basis of the enzyme, it would be more relevant to contrast with results of enzyme activity titers obtained by other authors, as it is written it is irrelevant. It is not a paper that pretends to describe the fundamentals of enzymatic reactions. The materials and methods section lack bibliographic support.

5.       line 323 indicates that yeast fermentation is a common method of purification, however, the statement is unfortunate since fermentation contains products of metabolism and enzymatic reaction as well as residual nutrients. Clarify and justify.

6.       In the digestive tract simulation experiments, what was the enzyme activity adjustment, i.e. how many units were adjusted and why? (U/mL or specific activity).

7.       In general, the paper is scarce on discussion with similar works and no shows the relevance of the research.

8.       The conclusion is necessary to refer to the objective declared in the introduction section.

9.       More than 50% of the references are outdated because this information shows old research scarce of novelty.

Reviewer 3 Report

The authors of MS foods-2287914 describe a research report focused on the production of long-chain IMOs by fermentation of maltose using B. subtilis strain AP-1. I suggest changing the title to "single-step" since the study was not (experimentally) designed with that objective and apparently being able to produce a series of IMOs during fermentation was serendipity. If not, the authors must improve the justification of use ""sngle-step".  A comparative description with other type or system of fermentation can to support it. 

The authors indicate they also evaluated the structural characteristics of the IMOs produced. This is not clear, what is tne meaning? no structural results are shown.

Round 2

Reviewer 2 Report

The manuscript has been revised enough.